# Single-cell RNA sequencing of healthy and diseased rat temporomandibular joint condyle cartilage

Sara Trbojevic[1,2], Xudong Dong[2], Robert Lafyatis[3], Michael S. Gold[4], Juan M. Taboas[1,2,5], Alejandro J. Almarza [1,2,5]*

1 Department of Bioengineering, Swanson School of Engineering, University of Pittsburgh, Pittsburgh, Pennsylvania, United States of America, 2 Center for Craniofacial Regeneration, University of Pittsburgh, Pittsburgh, Pennsylvania, United States of America, 3 Department of Medicine, University of Pittsburgh, Pennsylvania, United States of America, 4 Department of Neurobiology, University of Pittsburgh, Pittsburgh, Pennsylvania, United States of America, 5 Department of Oral and Craniofacial Sciences, School of Dental Medicine, University of Pittsburgh, Pittsburgh, Pennsylvania, United States of America

* aja19@pitt.edu

## Abstract

Osteoarthritis is a degenerative joint disease that disrupts the cellular homeostasis within cartilage tissues, promoting further disease progression that can lead to debilitating pain. Cartilage of the temporomandibular joint (TMJ) is unique among diarthrodial joints because they are of neural crest origin, rather than the mesoderm germ layer. TMJ cartilage also has different cellular architecture, where it is composed of a superficial fibrous layer, a proliferative layer, and a fibrocartilage layer overlying subchondral bone. Understanding of the cytopathological changes that occur during TMJ osteoarthritis (TMJOA) could potentiate therapies to mitigate disease progression and repair diseased tissues. Here, we analyzed the unique cell populations present in healthy and OA-induced condylar cartilage of adult rats through single-cell RNA-sequencing. TMJOA was established via our previous rat model to study the changes in the cellular composition of the condyle in response to OA. Several cell types could be uniquely identified, and the prominent matrix producing cells were fibroblasts and chondrocyte subsets. Our trajectory and pseudotime analysis revealed three cell fates stemming from a fibrochondrocyte-like population and two chondrocyte cell fates that stem from a shared progenitor population. We also found that Pleiotrophin is uniquely expressed in the proliferative zone by cells with a chondrocyte progenitor phenotype. In OA cartilage, differential gene expression in the fibroblast group revealed responses to inflammation, possibly through activation of chondrocyte differentiation. The chondrocyte group was highly metabolically active, indicative of rapid repair or remodeling. Cell-cell signaling analysis revealed that chondrocyte and chondrocyte progenitor communication became highly activated. Additionally, intracellular pathways that may contribute to cellular dysfunction and tissue remodeling were highly active, while pathways related to tissue catabolism appeared less active.

**Data availability statement:** All raw sequencing files (fastq) and processed files (rds) are available from the Gene Expression Omnibus (GEO) database (accession number GSE298882).

**Funding:** This work was supported by National Institute of Dental and Craniofacial Research (https://www.nidcr.nih.gov) with the following grants: R21DE027873-01A1 (awarded to AA, MG), R01DE030296 (awarded to AA, JT), and F31DE031967-01A1 (awarded to ST). The funder did not play any role in the study design, data collection and analysis, decision to publish, or preparation of the manuscript.

**Competing interests:** The authors have declared that no competing interests exist.

## Introduction

The temporomandibular joint (TMJ) is a complex, bilateral ginglymoarthrodial joint comprised of the mandibular condyle, articulating disc, and the glenoid fossa of the temporal bone. Temporomandibular disorders (TMDs) are a set of at least thirty different conditions, many with comorbidities and deep economic and psychosocial impact [1,2]. One of these conditions is TMJ osteoarthritis (TMJOA), which is a progressive degenerative joint disease that impacts the joint cartilage, subchondral bone, synovium, disc, and joint attachments. Hallmarks of TMJOA pathogenesis include cartilage degeneration, subchondral bone remodeling and thickening, development of osteophytes, loss of joint lubrication, and increased osteochondral angiogenesis [1]. While these changes have been described at the macroscopic level via standard histological techniques, changes at the cellular level have yet to be described in detail. This level of detail is important as it may suggest the cell type(s) primarily responsible for the pathological changes observed, if not those that could be supported to facilitate restoration of normal joint function.

Thus, the goal of this study was to characterize the cellular response to TMJOA-induced joints. Our incisor-splint OA-induction method was previously developed and validated to establish consistent cartilage and bone remodeling, including chondrocyte apoptosis, disrupted layer organization, and subchondral trabeculae corticalization and resorption [3]. Here, we used scRNA-seq on condyle cartilage tissues from healthy rats, and those in which TMJOA had been induced by altering joint loading with an our incisor splint [3,4]. We validated key markers of cell types using RNA-scope of lubricin (*PRG4*), collagen II (*COL2A1*), fibrillin 1 (*FBN1*), and pleiotrophin (*PTN*). We performed differential gene expression and pathway analyses comparing the experimentally splinted cartilage to the naïve cartilage from rat single-cell RNA-seq data. Across fibroblast and chondrocyte cell populations, our evidence suggests that OA was associated with an upregulation of pathways that may contribute to cellular dysfunction and tissue remodeling and a downregulation of pathways related to tissue catabolism.

## Materials and methods

### OA injury model

Thirty, 11-week-old female Sprague Dawley rats (Envigo, Indianapolis, USA) were used. The rats were housed in groups of two in an AAALAC accredited facility that is managed by the University of Pittsburgh's Division of Laboratory Animal Resources. All experimental procedures involving the animals were approved by the University of Pittsburgh Institutional Animal Care and Use Committee with protocol #20016559. The experimental group (n = 14) were subjected to an incisor splint to induce joint remodeling with histological signs of OA [4]. Briefly, rats were sedated with 4% isoflurane in an induction chamber, and once sedated, they continued to receive 2–3% isoflurane through a nose cone through the duration of the procedure. With their mouths held open, their upper incisors were cleaned with water and dried with cotton. The teeth were etched with 38% phosphoric acid (Etch-Rite etching gel, Pulpdent, USA)

for 30 seconds, rinsed again with water and dried with cotton. The teeth were then be primed with Adper Single Bond 2 (3M ESPE, Germany) and cured with an ultraviolet (UV) light for 20 seconds. Slanted (45° angle) stainless steel sleeves were then attached to the upper incisors with Filtek Z250 dental composite resin (3M ESPE, Germany) and cured with a UV light for 40 seconds. After placement, the rats were checked at least once a week for 4 weeks. Lower incisors were trimmed as necessary as they continuously grow in the rat. The non-splinted control group (n = 16) did not receive any procedures, and thus is a naïve control. No teeth trimming was necessary as they wear down on their own.

After the four week period, all the rats except for those used for micro-computed tomography(µCT) were euthanized with CO2 followed by cervical dislocation. Rats used for µCT were anesthetized using a mixture of Ketamine (55 mg/Kg), Xylazine (5.5 mg/Kg), and Acepromazine (1.1 mg/Kg) and then perfused transcardially with 80 ml 1X PBS followed by 80 ml of 4% Paraformaldehyde (PFA). Rats were divided as follows: 1) experimental (n = 8) and control (n = 8) for single-cell sequencing, 2) experimental (n = 3) and control (n = 3) for histology, 3) control (n = 2) for RNAScope, 4) experimental (n = 3) and control (n = 3) for µCT. Four weeks was chosen as a termination point because in our previous study we found consistent condyle remodeling, cartilage fibrillation, and cartilage cell apoptosis or disorganization [3,4].

## Histology and µCT

After euthanasia, the left and right TMJ condyles were dissected from each rat and the discs were removed (n = 3 rats per group). The following sample incubations were all performed at 4C with gentle agitation. The condyles were fixed in 15mL 10% formalin overnight and were then rinsed and decalcified in Immunocal (StatLab, USA) for one week. After decalcification, samples were rinsed and incubated in 15% sucrose overnight, followed by 30% sucrose overnight. Condyles were then mounted in OCT freezing medium in the anterior-posterior direction, flash frozen in liquid nitrogen, and stored at -80C until cryosectioning. Condyles were cryosectioned at 8µm, sections were mounted on microscope slides, and stored at -20C until histology was performed.

Prior to hematoxylin and eosin (H&E) staining, slides were chosen with sections from the same depth and screened for integrity under the microscope prior to staining. Slides were then allowed to reach room temperature and were stained with H&E. Images were taken at 4x magnification under brightfield illumination using a Nikon Eclipse microscope (Model TE2000-E, DS-Fi3 camera. Nikon, Japan).

After euthanasia, the rats (n = 3 per group) mandibular condyles were surgically exposed, and their discs were removed. Condyles were then fixed overnight in 4% PFA. Samples were rinsed in water and scanned for analysis on Scanco µCT 50 (Scanco Medical, Brüttisellen, Switzerland) system. 4.4 µm voxel size, 55KVp, 145µA intensity, 0.36 degrees rotation step (180 degrees angular range) and a 1000 ms exposure per view were used for the scans which were performed in 70% alcohol. The Scanco µCT software (HP, DEC windows Motif 1.6) was used for 3D reconstruction and viewing of images. After 3D reconstruction, volumes were segmented using a global threshold of 400 mg hydroxyapatite (HA) /mm³. It should be noted that no quantitative results were included due to the small sample size and therefore lack of statistical power to perform statistical analyses.

## Sample preparation and single-cell RNA-sequencing

After euthanasia, the left condyles were isolated, and their discs were removed. Condyle cartilage was sliced off the condyles using a #10 scalpel blade by carefully adjusting the blade angle to ensure bone was not removed with the cartilage. All cells from all n = 8 animals per group were pooled for analysis to ensure that enough cells were isolated and alive for analysis. Cartilage tissues were minced and then digested in 1 mg/mL Liberase DL (Sigma, USA) for 45 minutes, followed by enzyme inactivation via 2% fetal bovine serum (FBS) in PBS. The samples were centrifuged, supernatant was removed, and each group was resuspended in their respective (naïve and splinted) Cell Multiplexing Oligo (CMO) (10x Genomics ©) to tag naïve and splinted cells. After five minutes of CMO incubation at room temperature, samples were kept on ice for the remaining steps. Cells were washed three times with culture media containing 10% FBS. After washing,

cells were taken to the Single Cell Core for GEM generation and barcoding, cDNA amplification, library construction, and 3' gene expression (minimum read depth 20,000 read pairs/cell, paired-end, dual indexing) and cell multiplexing (minimum read depth 5,000 read pairs/cell, paired-end, dual indexing) sequencing following 10x Genomics Chromium Single Cell 3' Reagent Kits User Guide (v3.1 chemistry) [9]. It should be noted that because cells needed to be pooled and rats were not individually tagged, resulting in n = 1 per group, no statistical assessments between conditions were performed.

**scRNA-seq data processing**

FASTQ data files were uploaded to 10x Genomics Cloud Analysis software using the Cell Ranger Count c7.0.1 pipeline. Doublet removal, rat genome alignment (mRatBN7.1 Assembly), and count matrix generation was then completed. The resulting unfiltered feature matrix was loaded in RStudio (R v3.4.0). Data was filtered to utilize only features that were found in at least 3 cells and only cells that contained at least 100 features, resulting in 30,431 cells and stored in a Seurat Object utilizing R's Seurat library. Data was demultiplexed in two steps: 1) CellPlex Cell Multiplexing Oligo (CMO) barcodes were split using Seurat's *HTODemux* function, 2) CMO tags were assigned to the single cell data and added to the Seurat Objects metadata. The feature data was then filtered to remove cells with less than 100 features and more than 2,500 features This resulted in a final cell count of 25,288 cells. The updated Seurat Object was then exported and uploaded into Partek Flow software.

Data was further filtered using Partek Flow's *QA/QC* task and removing cells with high % mitochondrial counts (>20%) as they are likely damaged [10]. Using the *Filter Features* task, features that were not present in the remaining cells were excluded, resulting in 21,751 cells and 17,476 features. Data was then normalized using the *Normalization* task: 1) All gene raw read counts were divided by the number of counts per million in the sample (CPM), 2) added 1 and $Log_2$ transformed to deskew the data. Data were then filtered by CMO tag, excluding all cells that were not properly tagged, resulting in a final inclusion of 15,677 cells and 17,476 features. Data could then be filtered to include only fibrocartilage cells for the purpose of this study. This resulted in 4,257 cells from healthy rat fibrocartilage and 3,567 cells from splinted rat fibrocartilage (a total of 7,824 cells), which is on par with the expected yield because of both the low cellularity of cartilage when compared to other tissues and our rigorous filtering and exclusion criteria.

## Cell cluster generation and analysis

Dimensionality reduction was completed using the principal component analysis (*PCA*) task by using the top 2,000 features with the highest variance to calculate the top 100 principal components. Clusters were generated using the *Graph-based clustering* task on cells from both naïve and experimental animals. The Louvain clustering algorithm was used with a 0.6 resolution and structured based on 15 nearest neighbors (NN-Descent) and a Euclidian distance metric. A UMAP was generated using a neighborhood size of 30 and distance metric with a minimal distance metric of 0.3 to achieve the desired data dispersion. Cluster biomarkers were examined to manually identify and name cell clusters based on known gene expression markers that define phenotype. Using these defined clusters, key matrix-producing cell types were filtered for remaining analyses. Immune cells were left out of the primary analysis because rats were not perfused, and therefore, resident immune cells could not be distinguished from those circulating. To avoid potential misrepresentation of immune cell involvement in inflammation in our model, we instead focused on inflammatory mediation-related genes by tissue matrix-producing cells.

## Trajectory and pseudotime analysis

The data was used to run the Monocle 3 tool in Partek Flow, which maps likely branches of cell fate trajectories. The root node in the fibroblast trajectory was chosen as that with the highest *Dcn* expression, moderate *Col2a1* expression, and lowest *Prg4* expression, as this is the most classic fibroblast signature. Prg4 expression indicates a more specialized fibroblast state. The root node in the chondrocyte progenitor trajectory was chosen as that with the highest *Ptn* expression

and lowest *Alpl* expression, as this is representative of the least mature state. Pseudotime analysis was run, resulting in a graphical estimation of cell-state transitions.

### Enrichment analysis

Gene enrichment of each control group cluster compared to all other cluster control groups was performed by first determining the differentially expressed genes in each comparison using the Hurdle model. The differentially expressed gene sets were input into the PANTHER Classification System for GO enrichment analysis of biological processes, molecular functions, and cellular components.

### Differential gene expression and pathway analyses

Differential expression comparisons were done between the control and experimental samples within each cell cluster using the Hurdle model. Results were sorted by fold change to distinguish the most differentially expressed genes. Hierarchical clustering heatmaps were produced in Partek Flow. Differential analysis results were then used to complete pathway analyses for each cluster using Ingenuity Pathway Analysis, where pathways were sorted by p-value and fold change.

### RNAscope

RNAscope was performed using the 2.5 Duplex Detection Assay kit and probes from Advanced Cell Diagnostics (ACD). The probes chosen for RNA staining included collagen type II (*COL2A1*), pleiotrophin (*PTN*), fibrillin 1 (*FBN1*), and *PRG4*. We decided to use *COL2A1* as a primary chondrocyte marker, *PRG4* as a marker for the superficial layer of the fibroblast cluster, Fbn1 as another marker for the fibroblast cluster, and *PTN* as a marker to locate the population we called chondrocyte progenitor cell cluster. Formalin-fixed paraffin-embedding sample preparation was utilized for this assay. Briefly, TMJ condyles (n = 4) were collected from naïve rats (n = 2) after euthanasia, fixed in 10% neutral buffered formalin overnight, rinsed, demineralized, rinsed, and stored in 70% ethanol. Samples were processed for paraffin embedding in a tissue processor and embedded into paraffin blocks. Samples were sectioned in 5μm sections and mounted on slides. Before the assay, samples were deparaffinized, endogenous hydroxylase activity was blocked using RNAScope® Hydrogen Peroxide, and antigen retrieval/permeabilization using ACD custom pretreatment reagent was performed. The probes were then applied for two hours, washed, and stored in saline-sodium citrate buffer overnight. The next day, samples underwent two series of 5 RNA amplification steps, after which they were stained with the appropriate staining enzyme (HRP green or fast red). Samples were then counterstained with Hematoxylin, dried, cleared, and a coverslip was mounted. Images were taken at 20x magnification under brightfield illumination using a Nikon Eclipse microscope (Model TE2000-E, DS-Fi3 camera. Nikon, Japan). RNAscope signal was thresholded and converted to binary masks, and positive signal was quantified as percent area within manually defined anatomical regions of interest using Fiji (ImageJ). Semi-quantitative measurements were only performed on n = 1 sample per stain, so no statistics were performed and data should be used only as supportive evidence to the localization claims mentioned.

### Cell-cell signaling analysis

Cell signaling interactions were analyzed using the CellChat tool library in R. Cellchat objects were generated for both healthy and experimental datasets, and the communication probabilities between ligand-receptor pairs and pathways were inferred using 'computeCommunProb' and 'computeCommuneProbPathway'. Communication networks were aggregated using 'aggregateNet'. Both cellchat objects were then merged for comparison analysis using 'mergeCellChat'. Interactions were compared using 'compareInteractions' and were visualized using 'netVisual_heatmap'. Outgoing and incoming signaling changes were analyzed using 'netAnalysis_signalingRole_scatter' and 'netAnalysis_signalingRole_heatmap', and overall signaling pathway flow was calculated using 'rankNet'.

## Results

### TMJ condyles from splinted rats have disorganized, remodeled structure

Histological findings (Fig 1A) indicated clear morphological changes due to splinting within both the bone and cartilage. Condyles appear to be slanted or oval-shaped, consistent with robust bone remodeling. In addition, it appears that the bone has become more cortical with a loss of trabeculae. The cartilage layers appeared hypocellular, disorganized, and the surface appeared fibrillated. The typically distinct architecture of the cartilage layers had evident abnormalities, such as loss of superficial layer cells and the typical columnar structure that chondrocytes form. Histological results were

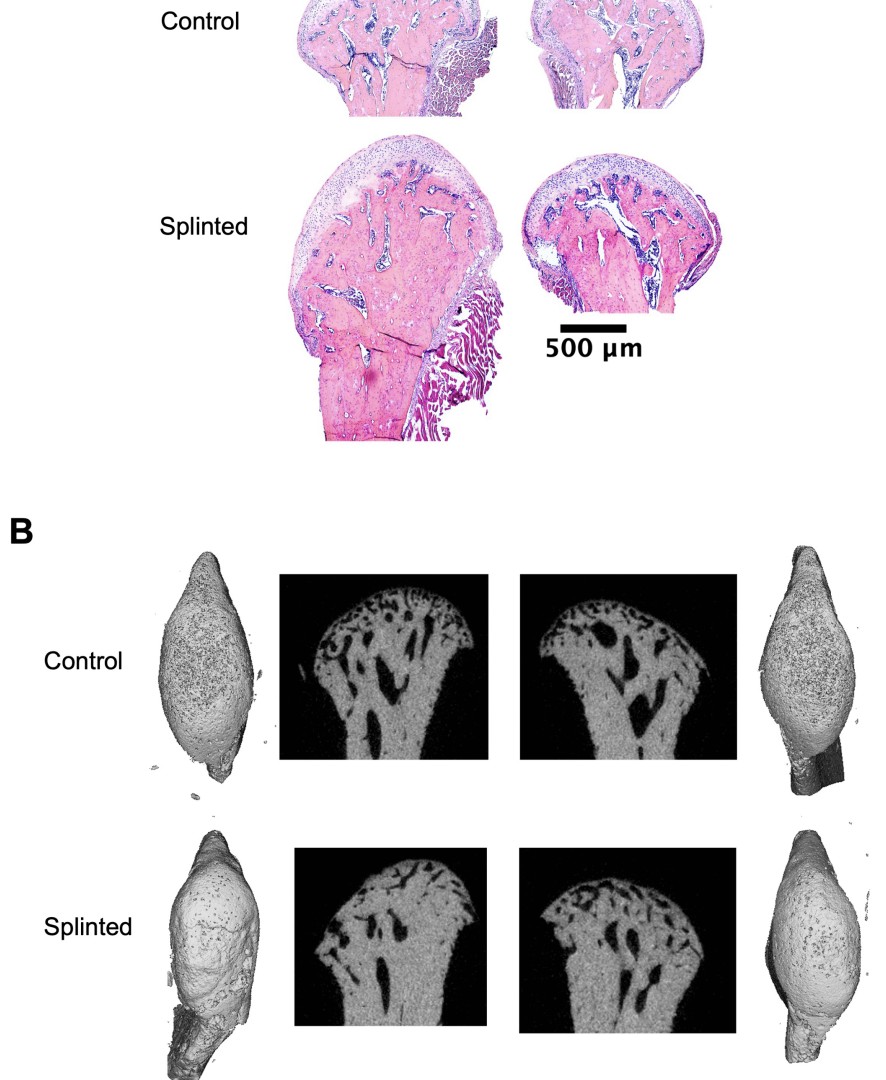

**Fig 1. Histology of representative non-splinted (top) and splinted (bottom) rat TMJ condyles, scale bar = 500µm (A). Micro-computed tomography 3D constructions and 2D sections of control and splinted condyles (B).**

consistent with prior findings [3]. µCT results further highlight the extensive bone remodeling in splinted rats compared to healthy. Healthy rats had a complex subchondral trabecular network with more trabecular structures, whereas the splinted group lost trabecular complexity, including loss of trabecular structures, trabecular thickening, and condyle surface corticalization (Fig 1B).

**scRNA-seq reveals subsets of matrix cells and immune cells in TMJ condyle cartilage**

As noted in Methods, the control rat cartilage (n = 8) resulted in a final count of 4,257 cells, and splinted rat cartilage resulted (n = 8) in a final count of 3,567 cells. Initial cell type classification resulted in two distinguishable groups: matrix cells and immune cells. Cells could be further divided into 12 clusters (Fig 2a,b), whose initial identities were determined based on expression of unique or well-established biomarkers of particular cell types (Fig 2c) [5,6]. Splinting had no detectable influence on the number of cells in each cell cluster, other than perhaps the red blood cells 1 cluster. Though, there were small increases in the percent of matrix cells, as well (Fig 2a).

Matrix cells (3,599 total) were comprised of fibroblasts (*PRG4*, *DCN*, *FBLN2*, *FBN1*, *COL1A1*), three chondrocyte clusters (*COL1A1*, *COL2A1*, *ACAN*, *SOX5/6/9*, *COL27A1*), and endothelial cells (*PECAM1*, *CDH5*). There are clear

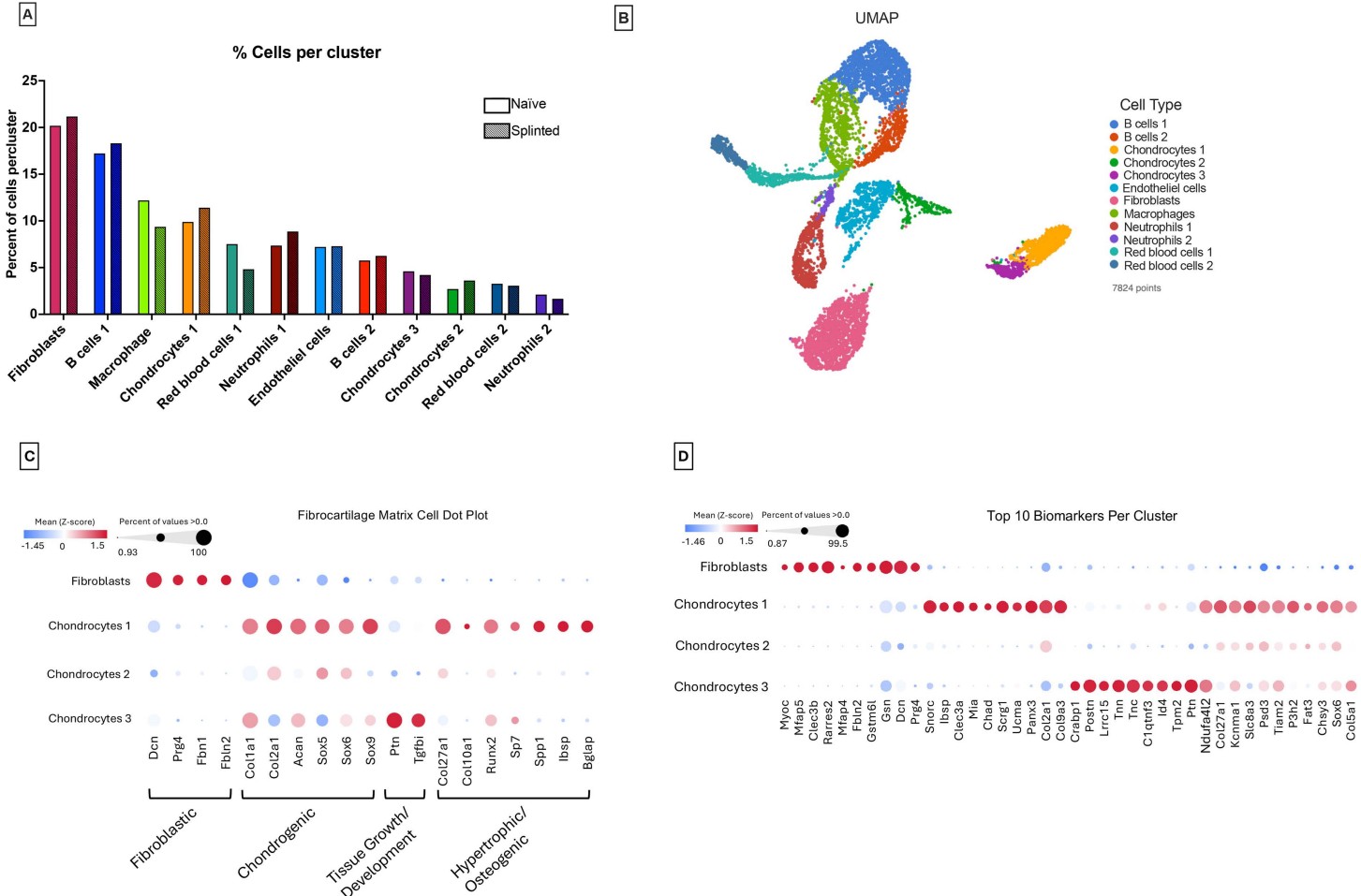

**Fig 2. Cell count in each cluster per group(A), Cell Cluster UMAP (B), Matrix cell biomarker dot plot (C), Dot Plot of the top 10 biomarkers in each cluster (D).**

distinctions among chondrocyte clusters, namely the expression levels or absence of chondrogenesis markers *COL2A1* and *SOX9*, developmental markers *PTN* and *TGFBI*, as well as osteogenic markers *SPP1*, *BGLAP*, and *SP7* (Fig 2c). These distinctions were further investigated for more detailed determination of the chondrocyte phenotypes they are likely representing.

Immune cell (*MYB*, *BACH2*, *CD79A*, *CD79B*, *S100A9*, *CEACAM1*, *LYZ2*), and red blood cell (*HEMGN*, *AHSP*) populations were identified. However, because rats were not perfused before sample collection, it is unclear whether immune cell populations were present in the joint tissues or if they were circulating. To complete our objective of analyzing fibrocartilage matrix cells, we filtered out immune cell, blood cell, and endothelial cell populations.

## Biomarker analysis of key cartilage cell subsets reveals fibroblast and chondrocyte subpopulations

To further characterize the results from the initial matrix gene expression analysis in fibroblasts, chondrocytes 1, chondrocytes 2, and chondrocytes 3, their top ten biomarkers were studied (Fig 2d), followed by trajectory analysis, RNAscope, and enrichment analyses to understand their role in cartilage matrix development, maintenance, and maturity.

Fibroblasts express genes associated with both fibroblasts and chondrocytes, though with lower expression, including *RARRES2*, *GSN*, and *DCN*, as well as the superficial layer marker *PRG4*. Chondrocytes 1 exhibit high expression of genes involved in chondrogenesis and chondrocyte maturity, including *SNORC*, *SCRG1*, *PANX3*, *COL2A1*, and *COL9A3*. These results indicate a maturing chondrocyte phenotype. Chondrocytes 2 have low expression of *GSN* and moderate expression of *COL2A1*, *COL27A1*, *PSD3*, and *SOX6*, all of which were more highly expressed in chondrocytes 1. This may indicate a transitional early-chondrocyte phenotype stemming from fibroblasts, which were more fibroblastic than chondrogenic (Fig 2c). Thus, we have renamed this cluster pre-chondrocytes (PreC). Chondrocytes 3 have high expression of *POSTN*, *TNC*, *ID4*, and *PTN*, genes expressed in early cartilage development or in chondrocyte progenitors; therefore, we have renamed them chondrocyte progenitors (ProC). Similar chondrocyte phenotypes have been reported previously [7–9].

## Trajectory and pseudotime analysis predicts the cell origins and subsequent differentiation states

To understand the course of gene expression changes across the four clusters of fibrocartilage depositing cells, a trajectory analysis and subsequent pseudotime calculation was performed, with the assumption that all cells in the same trajectories are from the same cell lineage. Trajectory analysis resulted in two trajectories: 1) fibroblasts & PreC, and 2) chondrocytes & ProC. The root node for trajectory 1 was chosen by analyzing overlapping gene expression of fibroblast-related genes (*DCN* & *PRG4*) and a chondrocyte-related gene (*COL2A1*). The node with *DCN*+/*COL2A1*+/*PRG4*- expression was chosen as the root node, while its branches had increased expression of *PRG4* and *COL2A1*, indicative of a less fibroblastic state. In the second trajectory, overlapping expression of a cartilage developmental gene (*PTN*), a chondrogenic gene (*ACAN*), and an osteogenic gene *ALPL* were analyzed. The node with *PTN*+/*ACAN*+/*ALPL*- expression was chosen as the root node, while its branches had higher *ACAN* and *ALPL*+ expression, indicative of a mature chondrocyte state (Fig 3a). Pseudotime results of trajectory 1 indicated a branching from a classical fibrochondrocyte state into two more branches with increasing fibroblast-like or chondrocyte-like expression through pseudotime. A third branch indicated the transition from the fibroblast cluster to the PreC cluster with a *COL2A1*+/*DCN*-/*PRG4*- final cell fate, supporting that the PreC cluster may stem from the fibroblast population rather than the ProC population. Pseudotime results of trajectory 2 indicated a branching from the *PTN*+ ProC cluster into two chondrocyte cell fates. Branches can be distinguished by the level of *ALPL* expression, with the branch with a higher pseudotime value having high *ALPL* expression (Fig 3b).

## RNAscope validates cell identity location in healthy rat TMJ condyle cartilage

To determine the location of the cells from some of the clusters, we performed RNAscope to localize gene expression patterns in situ. *COL2A1* RNA is primarily located in the hypertrophic cartilage layer. Biomarker expression of *COL2A1* was highest in the chondrocytes cluster, therefore it appears that these cells are primarily located in the hypertrophic cell layer,

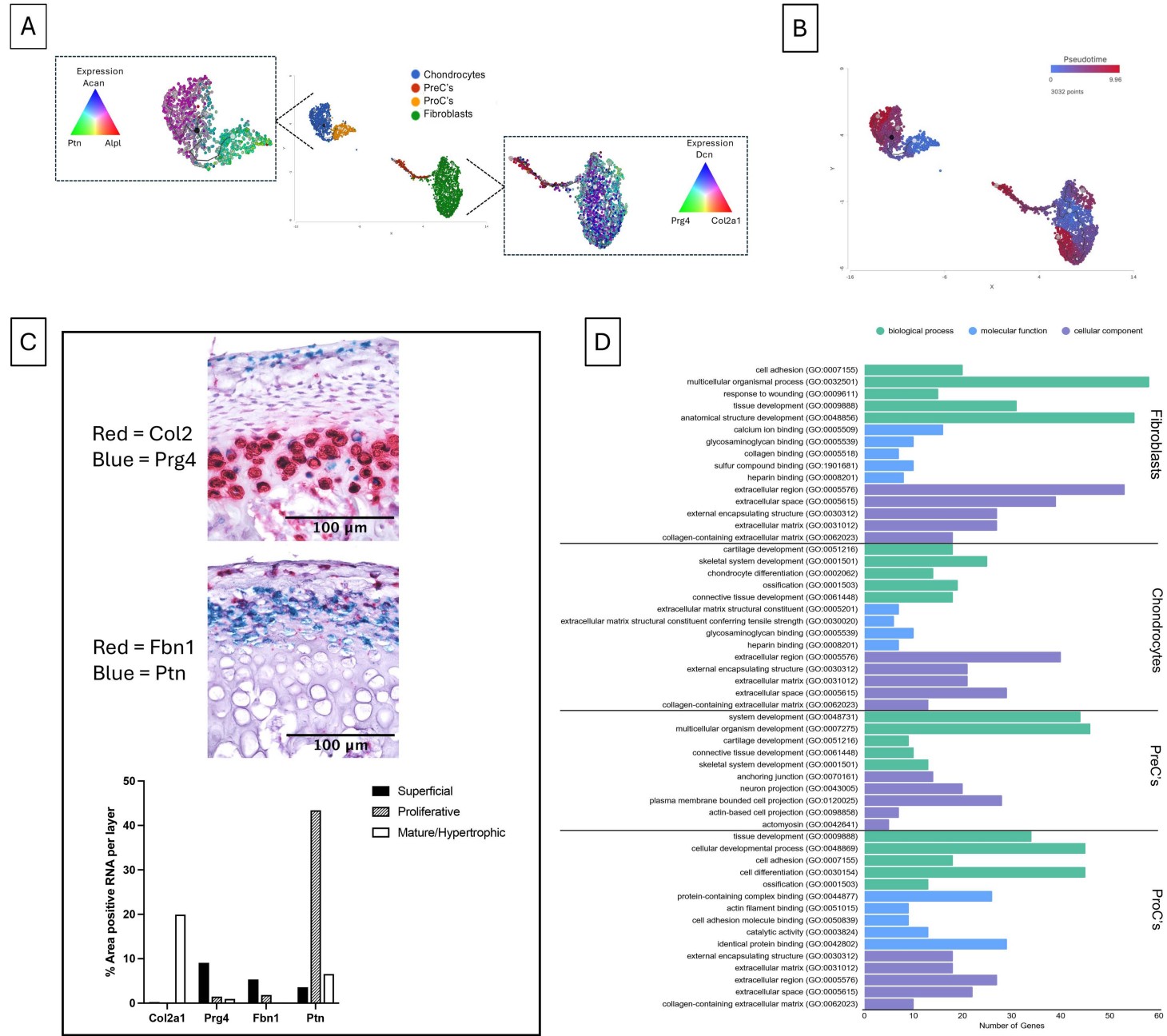

**Fig 3. Trajectory plot gene expression and cluster identity (A) and pseudotime plot(B).** RNAscope performed on condyle cartilage from healthy rats: stained for collagen 2 (*COL2A1*) in red and lubricin (*PRG4*) in blue (C, top), stained for fibrillin 1 (*FBN1*) in red and pleiotrophin (*PTN*) in blue (C, middle), and the percent area positive for each stain per layer (superficial, proliferative, and mature/hypertrophic) (n = 1) (C, bottom). Gene set enrichment results of each cluster, showing biological processes (green), molecular functions (blue), and cellular components (purple) (D).

with around 20% of the total area positive for *COL2A1*. Additional *COL2A1* expression underlying the superficial layer may indicate the location of the PreC group (Figs 2c, 3c). As anticipated, *PRG4* RNA was expressed in the superficial fibrous layer with around 10% of positive staining, verifying the expected identity of the fibroblast cluster. Notably, *PRG4* staining

was also detected in hypertrophic layer cells that express *COL2A1* (Fig 3c). *FBN1* RNA was stained most prominently in the superficial layer fibroblasts and in the underlying proliferative cell layer (Figs 2c, 3c). *PTN* RNA staining can be seen predominantly in the proliferative layer and had the highest amount of positive stain, around 45% area, across each gene and layer. *PTN* was highly expressed in the ProC cell group, consistent with the expression of this gene in cells undergoing differentiation. Notably, expression of *FBN1* and *PTN* in the proliferative layer suggests a shared pool of fibroblasts, PreC's, and ProC's (Figs 2c, 3c).

### Enrichment analysis identified distinctions in cellular behaviors between cell types

Gene set enrichment was used by comparing clusters to determine enrichment of: A) biological processes, B) molecular functions, and C) cellular components (Fig 3d), and findings from the top enrichment results of each category are summarized per cluster. Results indicated that fibroblasts are enriched for matrix production and structural integrity, including cell adhesion, tissue development, calcium ion binding, and glycosaminoglycan binding. Chondrocytes are enriched for maintaining and calcifying cartilage matrix, including cartilage development, chondrocyte differentiation, ossification, and glycosaminoglycan binding. PreC's are enriched for system development, cartilage development, anchoring junction, and neuron projection. Cellular component enrichment in PreC's was for cell motility and interaction, in contrast to the ECM production dominated by the other groups. This difference may support the migrating, transitional fibroblast/early-chondrocyte phenotype determined previously. ProC's are enriched for tissue development, cell differentiation, ossification, and protein-containing complex binding. Ossification enrichment likely signifies a subset of osteogenic progenitor cells, which is supported by the *PTN*+/*ALPL*+ progenitor cells in the trajectory graph (Fig 3a). Fibroblasts, chondrocytes, and ProC's all share enrichment for extracellular region components.

### Bite-shifting splints are associated with tissue damage and remodeling

Differential gene expression (DGE) and pathway analyses between splinted and control samples within each cluster were performed to determine the impact of splinting on cellular processes. Initially, the most upregulated and most downregulated differentially expressed genes (DEG's) were analyzed within each cluster for comparison. This analysis (Table 1) suggests that in the splinted group, fibroblasts seemed to have high fold changes in genes involved in inflammation (MATR3, GSTP1) and cytoskeletal response to mechanical stress (GEM), fatty acid metabolism regulation (NFIA) [10–13]. In the chondrocyte group (Table 2) we detected a marked metabolic response associated with increases in genes associated with chondrogenesis (SQLE, HMGCS1) and decreases in genes associated with osteogenesis and differentiation (KIF5B, BGLAP) [7–9,14]. PreC (Table 3) DEGs were implicated in inflammation, impaired differentiation, and catabolism (S100A8, CAMP, SAMD4A, ZFHX3, SKI), as well as a possible anabolic response (IGFBP5) [15–19]. Impaired differentiation likely prevents a phenotypic change from fibroblasts to chondrocytes, thereby resulting in a greater number of maturing chondrocytes. ProC gene expression (Table 4) was associated with increased cartilage catabolism, osteogenesis and Wnt signaling (NRCAM, STRN3, FZD8, LUM, GPX3), likely a reflection of increased cell death and loss of a chondrocyte phenotype associated with condyle remodeling [7,20–23].

Pathway analysis (Fig 4a) analysis between splinted and control rat cartilage was also performed. Top impacted pathways (sets of genes) relevant to our model (i.e., no cancer-related pathways, etc.) were considered for analysis, and here we summarize the results revealed by the largest significant (p < 0.05) increased and decreased pathways per cluster, most of which have been directly linked to OA. Results indicated notable pathological alterations in metabolism (AMPK, cholesterol biosynthesis, selenoamino acid metabolism, amino acid metabolism), tissue remodeling/degradation (RHO GTPase, *TGFβ* signaling, insulin receptor signaling, *VEGF*), and inflammatory stress responses (PPARα/RXRa activation, EIF Signaling, nonsense-mediated decay, eukaryotic translation). Compared to other cell groups, PreC's are most impacted by their diminished potential for regulation of gene expression and protein production, supporting the impaired

**Table 1. Top fibroblast DEGs and their functions based on expression state.**

| Fibroblasts | Gene | Expression State Function |
|---|---|---|
| **Overexpressed** | Matrilin3 (MATR3) | Increased ECM catabolism [10] |
| | GTP-binding protein (GEM) | Cytoskeleton reorganization [11] |
| **Under expressed** | Glutathione S-Transferase pi 1 (GSTP1) | Increased oxidative stress [12] |
| | Nuclear factor IA (NFIA) | Mitigated fatty acid metabolism imbalance [13] |

**Table 2. Top chondrocyte DEGs and their functions based on expression state.**

| Chondrocytes | Gene | Expression State Function |
|---|---|---|
| **Overexpressed** | Squalene Epoxidase (SQLE) | Involved in the cholesterol metabolic process, which is known to have key involvement in chondrocyte regulation [14,24] |
| | 3-hydroxy-3-methylglutaryl-CoA synthase 1 (HMGCS1) | Increased cholesterol synthesis [9] |
| **Under expressed** | Kinesin family member 5B (KIF5B) | Disruption of cytokinesis, leading to delayed terminal differentiation [8] |
| | Osteocalcin (BGLAP) | Repressed osteogenesis [7] |

**Table 3. Top PreC cell DEGs and their functions based on expression state.**

| PreC's | Gene | Expression State Function |
|---|---|---|
| **Overexpressed** | S100 Calcium Binding Protein A8 (S100A8) | Possibly increased inflammation, cartilage degeneration, and response to cytokines. S100A8 has also been associated with the presence of pain in knee joint OA [15] |
| | Insulin-like growth factor binding protein 5 (IGFBP5) | Increased anabolic response [25] |
| | Cathelicidin Antimicrobial Peptide (CAMP) | Increased inflammation, cartilage degeneration, and response to cytokines [16] |
| **Under expressed** | sterile alpha motif domain containing protein (SAMD4A) | Decreased chondrocyte differentiation [19] |
| | Zinc finger homeodomain 4 (ZFHX3) | Impaired chondrocyte differentiation [18] |
| | Ski proto-oncogene (SKI) | Inhibited chondrocyte differentiation [17] |

fibroblast to chondrocyte transition. ProC's have less expression of metabolic-related gene pathways, but rather, have upregulation of proliferation pathways. Furthermore, while not shown in Fig 4a, ProC's and chondrocytes had upregulation of OA pain-related pathways (NGF signaling pathways), though ProC's had the largest fold changes.

Aside from the unbiased approach of analyzing the top DEG's per cluster noted above, we were also curious about the impact of splinting on tissue marker genes (Fig 4b) and known OA-related genes from the OATargets database [26]

**Table 4. Top chondrocyte progenitor cell DEGs and their functions based on expression state.**

| ProC's | Gene | Expression State Function |
|---|---|---|
| **Overexpressed** | Neuronal cell adhesion molecule (NRCAM) | Involved in neural cell adhesion. Possibly upregulated by β-catenin [21] |
| | Striatin 3 (STRN3) | Possibly increased osteoblast differentiation under mechanical stress [22] |
| | Frizzled Class Receptor 8 (FZD8) | Activated non-canonical Wnt signaling [7] |
| **Under expressed** | Lumican (LUM) | Influenced cartilage degradation, reported to be upregulated in OA synovial fluid but downregulated in OA cartilage [20] |
| | Glutathione peroxidase 3 (GPX3) | Increased oxidative stress [23] |

(Fig 4c). Thus, we also performed a comparative analysis of these genes in cell clusters from splinted and control groups. Overall, differential gene expression analysis revealed evidence of cell differentiation and tissue remodeling playing roles in both OA progression and cartilage protection.

Fibroblasts appeared to have overexpression of lubrication and matrix markers (*PRG4*, *DCN*, *FBN1*, *FBLN2*, *LUM*) and overexpression of protective genes (*SOD2*, *SOD3, GRN*). In the chondrocyte cluster, markers of chondrocytes (*COL2A1*, *ACAN*), hypertrophic chondrocytes (*COL10A1*), and calcification of chondrocytes (*COL27A1*, *BGLAP*, *IBSP*, *PANX3*) were overexpressed. Chondrocytes also had overexpression of genes linked to the progression of osteoarthritis (*RUNX2*, *SP7*, *SPP1*, *COMP*) and those linked to suppressing osteoarthritis development (*THBS1*, *HIF1A*, *NFATC2*). In ProC's, tissue growth and cartilage development genes (*PTN*, *SOX6*, *SOX9*) and chondrocyte markers (*ACAN*, *COL1A1*) were overexpressed. Additional overexpression was seen in catabolic genes (*SP7*, *RUNX2*, *PGAM1*, *CTSK*) and possible anabolic genes (*TGFBI*, *TNC*, *EDIL3*). PreC's had overexpression of chondrocyte markers (*ACAN*, *COL1A1*, *COL2A1*, *COL9A1*, *COL11A1*), markers of hypertrophy and calcification (*SIK3*, *COL27A1*, *RUNX2*), and matrix catabolism (*S100A9*, *RUNX2*).

## Cell-cell signaling comparisons highlight key cellular players in stress response signaling

To further understand signaling roles of each cell type in both healthy and OA-induced cartilage, cell-cell signaling analysis using CellChat was conducted (Fig 5). Differential analysis was done to compare the number of interactions and interaction strength in OA versus control tissue, where an interaction is a ligand-receptor pair and interaction strength is the communication probability of interactions. Results indicated an increased number of interactions between all cell types, excluding PreCs. However, results showed increased interaction strength mainly between chondrocytes and ProCs. Generally, this highlights the increased number of signaling pathways and intensity of signaling between chondrocytes and ProCs, likely reflective of a shift towards progenitor activation during cellular stress (Fig 5a). Next, we identified and compared the information flow among the top signaling pathways in both experimental groups. As expected from a mechanoresponsive tissue with important structural properties, the highest signaling included Collagen, Thbs, Laminin, Ptn, and Fn1 pathways. Results further indicated that compared to OA tissue, healthy tissue was enriched for Ncam, App, Mif, and Ptprm signaling, indicative of cell adhesion, migration, and stress responsiveness that may have declined in OA [27–29]. In contrast, OA tissue was enriched for Periostin and Pcdh, and Adgre, which play roles in cartilage catabolism and mechanosensing (Fig 5b) [30,31].

When looking at incoming and outgoing signaling patterns (Fig 5c), it appears that PreCs were mostly involved in sending signals, of which remodeling-related pathways Ncam, Ptprm, and Bmp particularly declined in OA. It also seems that signaling to fibroblasts declined and instead shifted towards ProCs. In terms of emerging signaling patterns, early

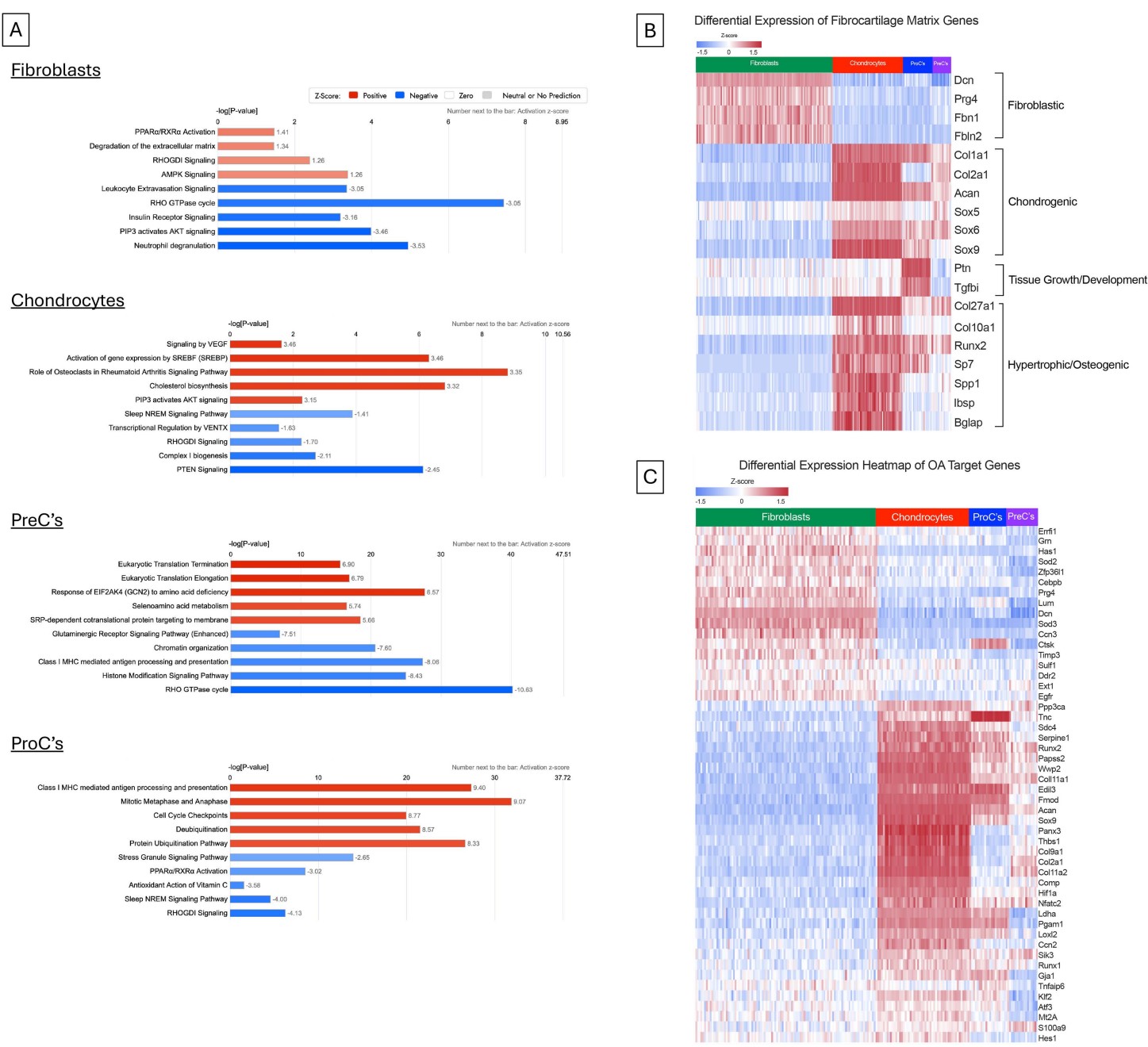

**Fig 4. Pathway analysis of top most upregulated and downregulated pathways of each cell type (splinted vs control).** Positive (red) z-score indicates upregulation in the splinted group relative to the control group (A). Differentially expressed fibrocartilage matrix genes (B) and differentially expressed genes from OATargets database (C).

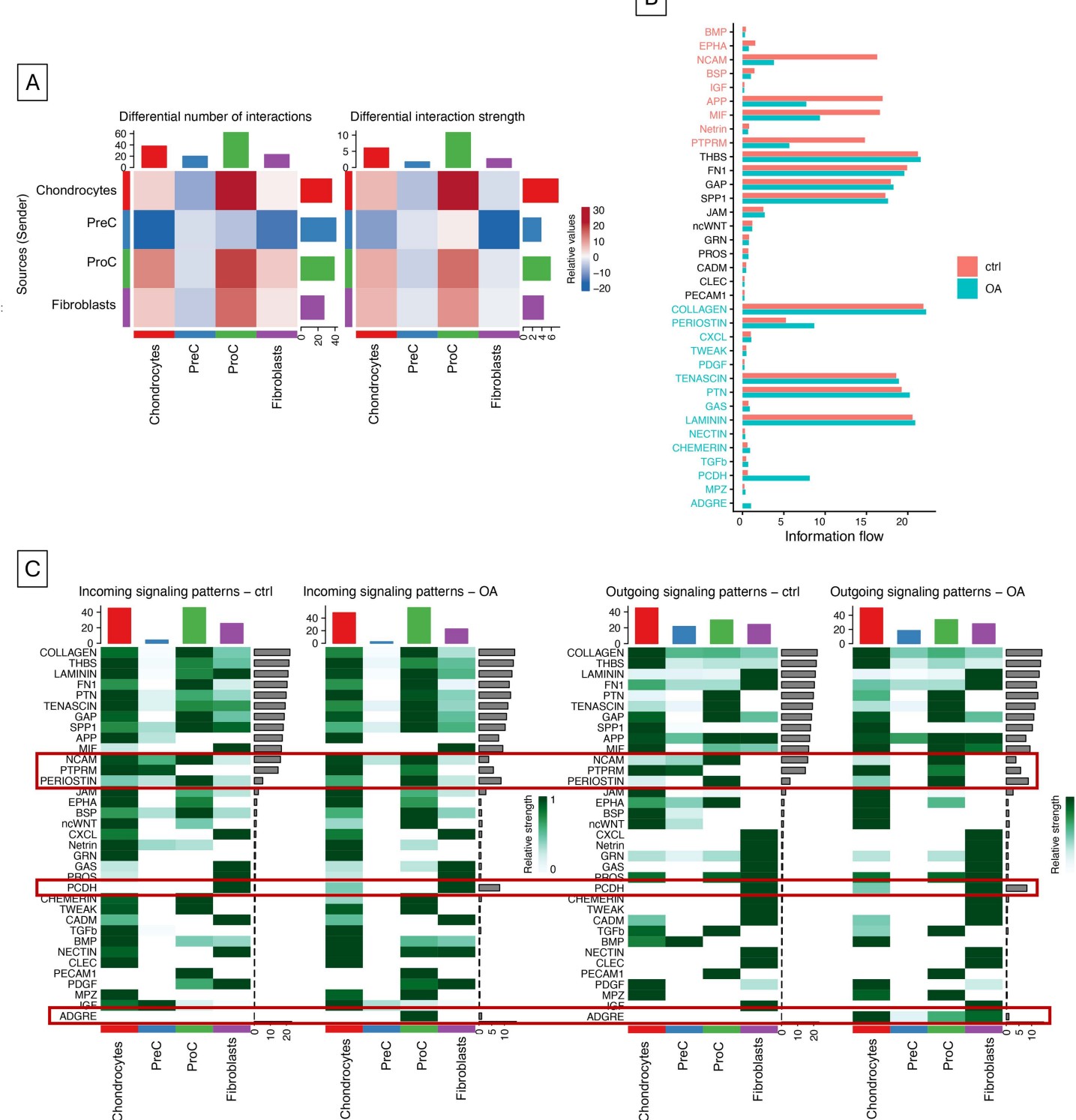

**Fig 5. Differentially expressed number of cell-cell interactions (ligand-receptor pairs) and interaction strength (communication probability), where red is overexpression in OA and blue is underexpression in OA(A).** Top communication signaling pathways among healthy control (red) and splinted OA (blue) cells (B). Incoming and outgoing signaling patterns of top pathways across cell types. Green is high communication strength

(probability) and white is low communication strength. The top bars indicate the total signaling from each cell, while the bars on the right indicate the total signaling of each pathway. Red boxes are highlighting the pathways with the most altered signaling strength.

chondrogenic-related chondrocyte Pcdh signaling, ProC Ptprm signaling, and Adgre signaling to ProCs emerged in OA. Top altered pathways are enclosed in boxes in Fig 5c.

## Discussion

In this study, we describe the cellular landscape of the TMJ mandibular condyle cartilage for the first time in both healthy and OA-induced rats. We characterize the cellular phenotypes and present the possible roles they play in cartilage homeostasis and stress response. Initial analysis of scRNA-seq data resulted in 12 distinct cell clusters, including fibroblasts, three chondrocyte groups, endothelial cells, red blood cells, and five immune cell groups. Because rats were not perfused, red blood cell and immune cell groups were excluded from further analysis, and fibrous and cartilage cell types were studied further. The four key subsets were identified as fibroblasts, chondrocytes, transitioning chondrocyte precursors, and chondrocyte progenitors.

In terms of cell clusters/population identities, fibroblasts were distinguishable due to their expression of known superficial layer ECM and lubrication genes (*DCN, PRG4, FBN1, FBLN2, COL1A1*), with in-situ RNAScope confirming their location in the superficial and proliferative layer. Chondrocytes expressed a range of genes present throughout all stages of chondrocyte maturation (*SCRG1, COL2A1, ACAN, COL10A1, PANX3, BGLAP*). In-situ analysis of *COL2A1* confirmed their location in the deeper cartilage layers. Chondrocyte precursors expressed similar genes as chondrocytes (*COL2A1, SOX5, SOX6, PSD3*), though at reduced levels, as they are likely in a lineage-primed state [32,33]. Trajectory and pseudotime analysis revealed that they transitioned from the fibroblast group into the early-chondrocyte transitional phenotype (Fig 3a, b). Chondrocyte progenitors exhibited an early developmental (*PTN, TGFBI, TNC*) phenotype, with suppression of MSC proliferation and in-situ localization to the proliferative/prehypertrophic cartilage region. Trajectory and pseudotime analysis positioned them in a distinct trajectory with chondrocytes. This separation of trajectories between fibroblasts-chondrocyte precursors and chondrocyte progenitors-mature chondrocytes were not interpreted as strict lineage restrictions, but rather grouping of similar transcriptional characteristics occurring along the same continuum in parallel. These findings support a model where fibroblasts gain transitional chondrogenic programming, while progenitor cells simultaneously commit to a cartilage lineage, both occurring in the shared proliferative zone. These cell populations support the known TMJ cartilage zones, as well as chondrocyte phenotypes that have been identified by others [32–35].

Differential gene expression analysis between splinted and naïve groups was consistent with the presence of tissue remodeling. This process was associated with the activation of AMPK, TGFb, and RHO GTPase pathways, as well as the emergence of a stress-responsive cell phenotype across all clusters. Many of the top altered pathogenic mediators described here have well-established links to human OA, and while the expression level in our early OA model may differ from that of human OA cartilage, it affirms the translatability of our model [36,37]. Specifically, dysregulated cholesterol and lipid metabolism, observed in most cell subsets, has been implicated in human OA cartilage degeneration and impaired chondrocyte functioning [38]. Altered AKT signaling has also been established in humans, though its exact role in OA remains unclear [39]. Elevation of RHO GTPase signaling gene, Rac1, has also been found in human OA, where it appears to accelerate hypertrophy and matrix degradation [40]. Regarding the cells identified here, fibroblasts showed signs of stress response with the upregulation of genes involved in ECM catabolism, reorganization, and oxidative stress. These changes were further supported by evidence of metabolic imbalance, suggesting an increased metabolic demand in the cartilage of splinted rats. One of the main markers of the fibroblast population is *PRG4*, which has been shown to be both increased and decreased in OA animal models. *PRG4* regulation appears to depend on the time of analysis (early/late remodeling), OA induction method, and the animal species. For example, *PRG4* expression has been shown to be

increased in early OA and decreased at later stages [41–45]. Consistent with previous data, increased *PRG4* expression in the present study is likely driven by altered mechanical loading, which stimulates *PRG4* transcription through activation of the CreB transcriptional regulator [41]. This response warrants further investigation at later time points to determine whether expression decreases as matrix degradation progresses [43].

Chondrocytes, in particular, appeared to be most impacted by splinting as evidenced by the pattern of DEGs which were associated with the disruption of cholesterol metabolism, which has been linked to MMP upregulation in OA-pathogenesis [46]. Our results align with prior studies showing that altered cholesterol metabolism occurs downstream of impaired hedgehog and TGFβ pathways, both of which are involved in chondrocyte regulation and bone formation [38]. Despite the expression of genes involved in chondrocyte maturation and cartilage calcification, it is notable that we detected decreased expression of genes linked to terminal differentiation and osteogenesis. While the occurrence of robust remodeling is indicated by histology, the presence of immature bone further suggests our model is capturing a state of rapid repair leading to impaired bone formation, which some studies suggest precedes cartilage degeneration [47]. Similar patterns of chondrocyte behavior have been observed in human knee OA, where a chondrocyte subset regulated homeostasis via metabolic response, similarly to the chondrocyte group in the current study [34]. The increased expression of genes associated with SREBF pathway activity, AKT signaling, and osteoclast activity is also consistent with changes associated with bone remodeling. Interestingly, in the knee OA study, the authors found that this subset of cells was primarily expressed in early osteoarthritis development, which is consistent with the structural changes observed in our OA model [4]. Taken together, despite the relatively long time-line used in the present study and the clear morphological evidence of joint remodeling, our results suggest that the changes driven by the splint at four weeks represent an early stage of OA, one in which the successful adaptation processes were still dominant.

The chondrocyte precursor cell group appear to be a subset of cells that are transitioning to a more chondrogenic phenotype from the fibroblast group. The transition appears to be impaired by splinting, evidenced by *SKI* downregulation and inflammatory gene upregulation, however, the cells are also playing a role in stress response shown by upregulation of EIF signaling and protein localization. Additionally, in DEG analysis, the magnitude of the changes were larger in chondrocyte precursor cells than any other cell group (Fig 4c). In other studies that performed scSeq on non-TMJ tissues (femoral cartilage, intervertebral disc), a similar group of cells has been found with both catabolic and anabolic responses, which may be stimulus-dependent, as evidenced by gene expression implicated in inflammation and impaired differentiation and protein production [34,35,48]. Impaired differentiation may inhibit the phenotypic change from a fibroblastic to chondrogenic state, leading to more mature chondrocytes in the largest chondrocyte population.

Chondrocyte progenitors appear to be a group of immature cells with impaired differentiation in response to stress from splinting, while also maintaining a pool of undifferentiated cells. This is evidenced by upregulation of gene expression and pathways involved in cell proliferation, protein ubiquitination, and Wnt signaling, but unlike the other cell groups, chondrocyte progenitors were less impacted by metabolic imbalance due to their lower metabolic requirements in an immature, less ECM-producing state [49,50]. As a result, they are more prone to apoptosis, supporting the magnitude of cell loss in the proliferative zone in our histology. While the cell type(s) responsible for OA pain has yet to be clearly defined, the elevated *NGF* pathway expression in chondrocyte progenitor cells is consistent with the emergence of pain drivers even at this early time point [51,52]. The link to *NGF* involvement in progenitor cell OA inflammation and impaired differentiation has been described previously, but this is the first time it has been linked on a single-cell level in the TMJ [53,54].

Cell-cell signaling analysis indicated that among the top differential signaling pathways included downregulation of Ncam, which has been tied to OA via the regulation of chondrocyte hypertrophy [27]. Upregulated signaling pathways included Periostin and Adgre, which have been linked to mechanically-induced cellular stress [30,31]. We also found the differential expression of signaling pathways not yet linked to OA, including Ptprm signaling, which shifted from and PreCs to ProCs in OA, and Pcdh, which emerged in chondrocytes in OA. Overall, broad signaling pattern shifts showed that the fibroblast-PreC lineage decreased signaling activity, particularly with remodeling-related pathways. In contrast, the

ProC-chondrocyte lineage became more active, engaging in increased remodeling-related signaling. This strongly implicates the chondrogenic cells as the primary regulators of the OA response through progenitor activation, though fibroblasts appear to be crucial in initial mechanosignaling communication.

While we were able to describe a number of interesting changes in gene expression in the cells of the TMJ condyle, it is important to acknowledge limitations of this study. First, cells were pooled from 8 rats to ensure sufficient yield and rats were not tagged individually. Therefore, biological variability could not be assessed, and statistical analyses between the healthy and splinted groups could not be performed. Thus, until further sequencing experiments are performed in a similar manner, results should be interpreted with caution. Pseudotime projections assume transcriptional differences reflect temporal progression, but in cartilage these differences may also arise from spatial organization, potentially confounding inferred trajectories. Separately, while we chose not to perfuse the rats studied to facilitate the recovery of condyle tissues, this precluded our ability to accurately analyze immune cell gene expression. Given the likely importance of these cells to response to tissue injury, analysis of this cell type would have helped provide context for the changes in gene expression detected on other cell types. However, non-immune cells initiate an immune response to inflammation via secretion of inflammatory mediators, such as cytokines, chemokines, and growth factors. Therefore, while the immune cell response is not captured in its entirety, their inflammatory recruitment cues have been captured. Inflammation-related results should be interpreted in regards to tissue-matrix cell inflammatory signaling, rather than the proceeding immune cell response. It should also be noted that chondrocytes produce pro-inflammatory factors during homeostatic dysfunction, even when overt inflammation is not present [55]. Finally, while used RNAScope for in situ RNA-level validation, it would be beneficial to perform protein-level expression validation in follow-up studies.

In our previous attempts at staining for inflammatory markers (TNFA, MMP13, INOS) we were unable to detect a difference between our splint model and naïve rats which may have been a reflection of the relatively early stage of OA progression as suggested by the results of the present study, as well as heterogeneity in the response to splinting. Nevertheless, we had hoped the higher resolution, unbiased approach of single-cell RNA sequencing would help understand inflammatory processes associated with splinting. Interestingly, the single cell results suggest that there are distinct cellular responses to the regulation of inflammation related genes. That is, pro-inflammatory gene expression were detected in all groups, though most prominently in fibroblasts and chondrocytes. While top DEG's of pre-chondrocytes included both pro- and anti-inflammatory gene expression, broad expression of inflammatory-related genes was lower than in other cell groups. This suggests that the transitional pre-chondrocytes are more protected from inflammatory-regulated catabolism. Nevertheless, it should be noted that more obscure markers of inflammation seemed to be more prominent across all cells, such as Matrilin 3 (*MATR3*), CathepsinK (*CTSK*), and S100 Calcium Binding Protein A8/A9 (*S100A8/A9*). These genes are thought to regulate inflammatory expression and induce a catabolic response to inflammatory cytokines. Thus, as previously noted, the changes observed at the four week timepoint in our splint model likely reflect an early stage of OA development where adaptive mechanisms appear to be still able to keep processes associated with overt tissue injury (cell death and inflammation) largely in check.

Overall, our single cell analysis of rat TMJ condyle cartilage revealed for the first time fibroblast and multiple chondrocyte phenotypes that appear to serve specific functions throughout cartilage cell maturation. Our results are consistent with the suggestion that the bite-shift splint drives the development of early stage TMJOA associated with ECM organization and the upregulation of pro-inflammatory genes in ECM cells. Given the wide number of changes observed across multiple cell types, our results are also consistent with the suggestion that ECM cells are highly responsive to changes in the local environment and play a critical role in the initiation of adaptive responses to altered loading. Finally, this work provides the first insight into the single-cell landscape of TMJ cartilage from both healthy and mechanically-induced TMJOA cartilage in a rat model, and provides new targets to study disease progression and potential therapeutics.

## Acknowledgments

The authors would like to thank the University of Pittsburgh Single Cell Core research staff for conducting our sequencing experiment and initial data analysis. We would also like to thank the Center for Craniofacial Regeneration micro-CT core for conducting micro-CT reconstructions of our condyle samples.

## Author contributions

**Conceptualization:** Sara Trbojevic, Michael S. Gold, Alejandro Almarza.

**Data curation:** Sara Trbojevic.

**Formal analysis:** Sara Trbojevic.

**Funding acquisition:** Sara Trbojevic, Michael S. Gold, Juan M. Taboas, Alejandro Almarza.

**Investigation:** Sara Trbojevic, Xudong Dong.

**Methodology:** Sara Trbojevic, Xudong Dong, Robert Lafyatis, Michael S. Gold, Alejandro Almarza.

**Project administration:** Sara Trbojevic, Xudong Dong, Alejandro Almarza.

**Resources:** Robert Lafyatis.

**Supervision:** Robert Lafyatis, Michael S. Gold, Juan M. Taboas, Alejandro Almarza.

**Validation:** Sara Trbojevic, Xudong Dong, Robert Lafyatis.

**Visualization:** Sara Trbojevic.

**Writing – original draft:** Sara Trbojevic.

**Writing – review & editing:** Sara Trbojevic, Xudong Dong, Robert Lafyatis, Michael S. Gold, Juan M. Taboas, Alejandro Almarza.

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
