## [Decision Letter · Decision Letter 0]

9 Dec 2025

Dear Dr. Almarza,

Thank you for submitting your manuscript to PLOS ONE. After careful consideration, we feel that it has merit but does not fully meet PLOS ONE’s publication criteria as it currently stands. Therefore, we invite you to submit a revised version of the manuscript that addresses the points raised during the review process.

We look forward to receiving your revised manuscript.

Kind regards,

Esmaiel Jabbari, PhD

Academic Editor

PLOS One

Journal Requirements:

Reviewers' comments:

Reviewer's Responses to Questions

**Comments to the Author**

1. Is the manuscript technically sound, and do the data support the conclusions?

Reviewer #1: Yes

Reviewer #2: No

Reviewer #3: Partly

2. Has the statistical analysis been performed appropriately and rigorously?

Reviewer #1: Yes

Reviewer #2: No

Reviewer #3: No

3. Have the authors made all data underlying the findings in their manuscript fully available?

Reviewer #1: Yes

Reviewer #2: No

Reviewer #3: Yes

4. Is the manuscript presented in an intelligible fashion and written in standard English?

Reviewer #1: Yes

Reviewer #2: Yes

Reviewer #3: Yes

Reviewer #1: 1. The authors state that cells from all n=8 animals per group were pooled for analysis to ensure sufficient cell yield. This is a significant limitation, as it precludes the assessment of biological variability between individuals and, crucially, any formal statistical testing for differences between the naïve and splinted conditions (e.g., using a mixed-model approach accounting for multiple biological replicates). This critical limitation must be explicitly stated and emphasized in the Discussion, and the conclusions should be framed with appropriate caution. We recommend explicitly mentioning this design choice and its implications for statistical inference in the main text (e.g., Methods or Results) and the Discussion.

2. Due to the lack of perfusion, immune cell populations were excluded from analysis as they were considered potentially circulatory. However, tissue-resident immune cells (e.g., macrophages in the synovium) play a key role in OA. This issue should be more thoroughly addressed by (1) explicitly stating this as a limitation in the Methods and (2) providing a more in-depth discussion of how this decision might impact the interpretation of the inflammatory microenvironment in OA.

3. The filtering criteria for "fibrocartilage cells" need clearer description. What specific marker genes or clustering results informed this selection? Furthermore, details of the trajectory inference, specifically the process and exact criteria for selecting the root nodes, should be elaborated upon to ensure the analysis is reproducible.

4. Given the identification of multiple cell types, leveraging the existing scRNA-seq data to infer potential intercellular communication changes in OA (using tools like NicheNet or CellChat) could provide deeper insights into the pathological mechanisms. This could be a valuable addition for future consideration if feasible.

5. The Discussion could briefly contextualize how the cell subsets and key pathways identified (e.g., cholesterol metabolism, NGF signaling) in this rat TMJOA model compare or contrast with known pathways in human TMJOA or peripheral joint OA, enhancing the potential translational relevance of the findings.

Reviewer #2: The manuscript analyzed the unique cell populations present in healthy and OA-induced condylar cartilage of adult rats through single-cell RNA-sequencing. However, the analysis

is too rough and need to do further and deeper experiments.

1. Figure 1 need add further experiments, such as S.O. staining, Micro-CT images and Quantitative analysis.

2. Single cell analysis figures can put together to be one figure. Further experiments such as Westernblot verification for OA Target Genes should be done.

Reviewer #3: This study presents the first single-cell landscape of the rat TMJ condyle in both health and a mechanically induced OA model. The identification of "Pre-chondrocytes" (PreC) and "Chondrocyte Progenitors" (ProC) provides valuable insight into the unique cellular architecture of the mandibular condyle, particularly given its neural crest origin. The manuscript is generally well-written and the trajectory analysis offers a compelling hypothesis regarding lineage plasticity. However there are a number of major and minor concerns, outlined below:

1. The "ProC" Population and Developmental Context: The authors identify a "Chondrocyte Progenitor" (ProC) population marked by Ptn, Tnc, and Id4. In the trajectory analysis (Figure 4), ProC appears as a distinct root separate from the fibroblast lineage.

Critique: Standard developmental models of the TMJ suggest that the superficial polymorphic/proliferative layer (often called the reserve zone) contains the progenitors for the underlying cartilage. This layer is usually continuous with the fibrous zone.

Action: Please clarify: Does the ProC population correspond spatially to the "polymorphic layer" in histology? The RNAscope for Ptn (Figure 5B) shows expression in the proliferative layer. The authors should discuss whether the ProC and Fibroblast clusters represent a continuum of the superficial zone or distinct lineage restrictions

2. Clarification of "Inflammation" vs. "Immune Response": The authors excluded immune cells because the rats were not perfused. However, the results section extensively discusses inflammatory pathways in matrix cells (e.g., Fibroblasts expressing Matr3, PreC expressing S100a8).

Critique: It is crucial to distinguish between intrinsic inflammatory signaling by chondrocytes (which drives catabolism) and extrinsic immune cell infiltration.

Action: Please explicitly discuss that the "inflammatory response" observed is likely an autocrine/paracrine stress response of the matrix cells themselves, rather than immune infiltration, as the immune cells were bioinformatically removed.

3. Validation Quantification: The RNAscope images (Figure 5) are excellent for localization. However, to bolster the claims of cell identity, quantification (even semi-quantitative counting of positive cells in specific zones) would strengthen the link between the scRNA-seq clusters and the histological layers.

Minor Revisions

Abstract: The phrase "they are of neural architecture" is awkward. Consider revising to "they are of neural crest origin."

Methods - Sample Pooling: Please clarify if the "pooling" involved barcoding individual animals before pooling (hashing) or if they were physically pooled into one suspension without individual tags. The text mentions "Cell Multiplexing Oligo (CMO)... to tag naïve and splinted cells", implying two pools (one per group), not individual animal tagging. This confirms n=1 per group. Please be precise here.

Figure 7 (Pathways): The bar charts show "Activation z-score" and "-log(p-value)". Ensure the legend clarifies that "Positive" z-score means upregulated in the Splinted group relative to Control.

Typos:

: "The cartilage layers, appeared hypocellular..." (Remove comma). : "The transition is appears to be..." (Remove "is")

.

Reviewer #1: No

Reviewer #2: No

Reviewer #3: **Yes:** Neal AnthwalNeal AnthwalNeal AnthwalNeal Anthwal

---

## [Author Response · Author response to Decision Letter 1]

19 Jan 2026

We would like to thank the reviewers for their thoughtful comments that have led to an improved manuscript revision. We specifically addressed each one, including clarifying methods and limitations. We have also added additional analyses, including cell-cell signaling analysis using CellChat, as well as a semiquantitative analysis of RNAScope results. All responses are copied below, but are also included in the Response to Reviewers file in this resubmission.

Reviewer #1:

1. The authors state that cells from all n=8 animals per group were pooled for analysis to ensure sufficient cell yield. This is a significant limitation, as it precludes the assessment of biological variability between individuals and, crucially, any formal statistical testing for differences between the naïve and splinted conditions (e.g., using a mixed-model approach accounting for multiple biological replicates). This critical limitation must be explicitly stated and emphasized in the Discussion, and the conclusions should be framed with appropriate caution. We recommend explicitly mentioning this design choice and its implications for statistical inference in the main text (e.g., Methods or Results) and the Discussion.

This limitation has now been addressed in both the methods section and the discussion to ensure transparency of the design and lack of statistical power/analyses.

2. Due to the lack of perfusion, immune cell populations were excluded from analysis as they were considered potentially circulatory. However, tissue-resident immune cells (e.g., macrophages in the synovium) play a key role in OA. This issue should be more thoroughly addressed by (1) explicitly stating this as a limitation in the Methods and (2) providing a more in-depth discussion of how this decision might impact the interpretation of the inflammatory microenvironment in OA.

We have now included this limitation in the methods and further addressed it in the discussion by clarifying that while immune cell-specific response could not be accurately captured, tissue-matrix cell inflammatory signaling via cytokine, chemokine, and growth factor secretion was captured and interpreted as such.

3. The filtering criteria for "fibrocartilage cells" need clearer description. What specific marker genes or clustering results informed this selection? Furthermore, details of the trajectory inference, specifically the process and exact criteria for selecting the root nodes, should be elaborated upon to ensure the analysis is reproducible.

This description was already present in the results section, but it has now been expanded upon in the methods section, describing specific genes and their expression levels. Briefly, high Dcn, moderate Col2a1, and low Prg4 was chosen as the fibroblast root node, as this is the most classic fibroblast signature. High Ptn and low Alpl was chosen as the chondrocyte progenitor root node, as this indicates the least mature chondrocyte gene expression.

4. Given the identification of multiple cell types, leveraging the existing scRNA-seq data to infer potential intercellular communication changes in OA (using tools like NicheNet or CellChat) could provide deeper insights into the pathological mechanisms. This could be a valuable addition for future consideration if feasible.

CellChat communication analysis has been added, highlighting a decline in PreC signaling and a shift towards increased chondrocyte and ProC signaling. We also identified key pathways that were altered in our OA-induced model.

5. The Discussion could briefly contextualize how the cell subsets and key pathways identified (e.g., cholesterol metabolism, NGF signaling) in this rat TMJOA model compare or contrast with known pathways in human TMJOA or peripheral joint OA, enhancing the potential translational relevance of the findings.

The suggested addition to the discussion has been added, providing links from the top altered pathways in our data to those that are well-established in human OA. It should be noted that the activation or inactivation of the pathway in our model may differ from that of human OA, likely due to the stage of disease development that we are modeling. However, the inter-species similarities remains an important addition.

Reviewer #2:

1. Figure 1 need add further experiments, such as S.O. staining, Micro-CT images and Quantitative analysis.

Micro-CT 3D reconstructions and sections of the control and splinted condyles have been added to figure 1, along with their acquisition methods. These images further highlight the extensive remodeling and corticalization changes in the bone due to splinting, which is supported by the single-cell data.

2. Single cell analysis figures can put together to be one figure. Further experiments such as Westernblot verification for OA Target Genes should be done.

Some figures have now been combined, resulting in 5 figures rather than 9. To the second point, while we acknowledge the importance of protein-level expression validation, the primary objective of this study was to define the cell type specific transcriptomic profiles and how these profiles change in our splinted rat model. This body of work can then serve as a foundation for the development of more pointed, protein-level studies. However, we have added this as to the limitations in our discussion.

Reviewer #3:

1. The "ProC" Population and Developmental Context: The authors identify a "Chondrocyte Progenitor" (ProC) population marked by Ptn, Tnc, and Id4. In the trajectory analysis (Figure 4), ProC appears as a distinct root separate from the fibroblast lineage.

Critique: Standard developmental models of the TMJ suggest that the superficial polymorphic/proliferative layer (often called the reserve zone) contains the progenitors for the underlying cartilage. This layer is usually continuous with the fibrous zone.

Action: Please clarify: Does the ProC population correspond spatially to the "polymorphic layer" in histology? The RNAscope for Ptn (Figure 5B) shows expression in the proliferative layer. The authors should discuss whether the ProC and Fibroblast clusters represent a continuum of the superficial zone or distinct lineage restrictions

The trajectory analysis is interpreted as groups with similar transcriptional programming maturing along parallel paths simultaneously, with both PreC and ProC populations stemming from the proliferative zone. This clarification has been expanded upon and added to the discussion.

2. Clarification of "Inflammation" vs. "Immune Response": The authors excluded immune cells because the rats were not perfused. However, the results section extensively discusses inflammatory pathways in matrix cells (e.g., Fibroblasts expressing Matr3, PreC expressing S100a8).

Critique: It is crucial to distinguish between intrinsic inflammatory signaling by chondrocytes (which drives catabolism) and extrinsic immune cell infiltration.

Action: Please explicitly discuss that the "inflammatory response" observed is likely an autocrine/paracrine stress response of the matrix cells themselves, rather than immune infiltration, as the immune cells were bioinformatically removed.

This comment has been addressed in another similar comment (review 1, #2). We have now included the exclusion of immune cells more explicitly in the discussion limitations, as well as a discussion on the role of matrix cells in the inflammatory response.

3. Validation Quantification: The RNAscope images (Figure 5) are excellent for localization. However, to bolster the claims of cell identity, quantification (even semi-quantitative counting of positive cells in specific zones) would strengthen the link between the scRNA-seq clusters and the histological layers.

Semi-quantitative measurements of positive stain in each layer (reported as %area) have been added, including a bar graph of the results. However, with a measurement sample size of n=1 per stain, no statistics was added and it has been mentioned in the methods that these measurements are only to be viewed as additional supportive evidence for the qualitative claims of RNA localization.

Minor Revisions

Abstract: The phrase "they are of neural architecture" is awkward. Consider revising to "they are of neural crest origin."

Methods - Sample Pooling: Please clarify if the "pooling" involved barcoding individual animals before pooling (hashing) or if they were physically pooled into one suspension without individual tags. The text mentions "Cell Multiplexing Oligo (CMO)... to tag naïve and splinted cells", implying two pools (one per group), not individual animal tagging. This confirms n=1 per group. Please be precise here.

This has now been addressed and clarified in the methods.

Figure 7 (Pathways): The bar charts show "Activation z-score" and "-log(p-value)". Ensure the legend clarifies that "Positive" z-score means upregulated in the Splinted group relative to Control.

This clarification has been added to the figure caption.

Typos:

: "The cartilage layers, appeared hypocellular..." (Remove comma). : "The transition is appears to be..." (Remove "is")

The mentioned errors have been fixed.

---

## [Decision Letter · Decision Letter 1]

14 Apr 2026

Single-cell RNA sequencing of healthy and diseased rat temporomandibular joint condyle cartilage

PONE-D-25-53506R1

Dear Dr. Almarza,

We’re pleased to inform you that your manuscript has been judged scientifically suitable for publication and will be formally accepted for publication once it meets all outstanding technical requirements.

Kind regards,

James J Cray Jr., Ph.D.

Academic Editor

PLOS One

Additional Editor Comments (optional):

Reviewers' comments:

Reviewer's Responses to Questions

**Comments to the Author**

Reviewer #1: All comments have been addressed

Reviewer #3: All comments have been addressed

2. Is the manuscript technically sound, and do the data support the conclusions?

Reviewer #1: Yes

Reviewer #3: Yes

3. Has the statistical analysis been performed appropriately and rigorously?

Reviewer #1: Yes

Reviewer #3: I Don't Know

4. Have the authors made all data underlying the findings in their manuscript fully available?

Reviewer #1: Yes

Reviewer #3: Yes

5. Is the manuscript presented in an intelligible fashion and written in standard English?

Reviewer #1: Yes

Reviewer #3: Yes

Reviewer #1: The authors have answered all of my questions of this article, and I am satisfied with the authors' responses.

Reviewer #3: The Reviewers have answered the comments to a suffient standard that the article is now suitible for publication in PLOSOne

.

Reviewer #1: No

Reviewer #3: **Yes:** Neal AnthwalNeal AnthwalNeal AnthwalNeal Anthwal

---

## [Editor Report · Acceptance letter]

PONE-D-25-53506R1

PLOS One

Dear Dr. Almarza,

I'm pleased to inform you that your manuscript has been deemed suitable for publication in PLOS One. Congratulations! Your manuscript is now being handed over to our production team.

Kind regards,

on behalf of

Dr. James J Cray Jr.

Academic Editor

PLOS One